# THOUGHTBUBBLES: AN UNSUPERVISED METHOD FOR PARALLEL THINKING IN LATENT SPACE

## ABSTRACT

Current approaches for scaling inference-time compute in transformers rely on training them to emit explicit chain-of-thought tokens before producing an answer. While these methods are powerful, they are limited because they cannot be applied during pretraining and are limited to only serially-generated, natural-language verbalization to scale inference-time compute. In this work, we propose **Thoughtbubbles**, a transformer variant that natively performs parallel adaptive computation in latent space by learning to fork or delete residual streams. Thus, tokens that require a large amount of computation can form a "bubble" of cloned residuals in the middle of the network for additional computation. Crucially, this behavior is learned during pretraining with only language modeling loss. **Thoughtbubbles** outperforms both standard decoder LMs as well as non-adaptive parallel computation approaches on `OpenWebText` and `peS2o` perplexity and in zero-shot evaluations such as HellaSwag and LAMBADA after pretraining across 150M to 772M parameter scales. The implicit nature of our method enables adaptive computation to be learned starting at pretraining time, paving the way to unify train-time and test-time scaling behaviors.

## 1 INTRODUCTION

Despite their unprecedented success, Transformers (Vaswani et al., 2017) only have a fixed computation budget and working memory, which present both a theoretical (Merrill & Sabharwal, 2023) and practical limit (Sanford et al., 2024) for solving complex, multi-step problems.

Due to the growing interest in extending the capabilities of transformers for difficult multi-step problems, many efforts are underway to surpass this bounded-computation limitation of transformers. The earliest and simplest is Chain of Thought (CoT) (Wei et al., 2023), where a transformer language model is explicitly prompted to provide a set of reasoning steps. This technique allows the model to break a problem down to subproblems, solve them individually, and cache intermediate results for the full solution—enabling a simple form of problem adaptivity (Merrill & Sabharwal, 2024).

Extending this result, Pfau et al. (2024) show both theoretically and practically that CoT improves the expressiveness of transformers—even when the CoT traces are replaced with a unique thinking token (dots) at test time: indicating that even residual streams alone, not serial recurrence, can improve computational performance.

Such an insertion of additional residual streams, the so-called "pause tokens", has since become a growing trend of recent architecture research. Though methods vary in terms of where to actually insert the thinking tokens(Herel & Mikolov, 2024; Sun et al., 2025; Goyal et al., 2024), all pause token approaches insert additional computation streams prior to computation—limiting the model's ability to allocate intermediate computation that is useful only in some, but not all layers (e.g., computation which is useful only after a few layers of attention). As Sun et al. (2025) notes, determining the location of these pause tokens often requires manual design following the structure of the problem, which may be intractable for general language models.

In response, we introduce **Thoughtbubbles**, a novel Transformer-based architecture which enables the unsupervised and dynamic allocation of additional parallel residual streams for extra computation and memory. We achieve this by introducing a novel forking mechanism between some layers,

which computes and maintains a cumulative score for every residual stream and uses it to decide whether to *create* extra residuals or to *delete* existing ones.

This formulation makes dynamic computation a budget-bounded allocation problem of these scores. In order to train these scores to be useful, we use these scores to mask both the model's ability to attend to residual streams with low scores as well as limit the model's ability to update them at each layer. This attenuation forces the model to provide higher scores to residual streams it deems more important, which will also result in increased forking of those streams. At the end of encoding, our model will decode each residual streams separately, including forked ones. Weighted by their scores, streams for the same token are then averaged together to produce the final distribution.

Thus, our approach will essentially create "bubbles" of latent computation consisting of forked residuals for difficult tokens (i.e., those with high cumulative scores) for additional thinking, before merging them to produce the final output token.

We conduct a variety of pretraining experiments across 150M to 772M scales and make the following contributions:

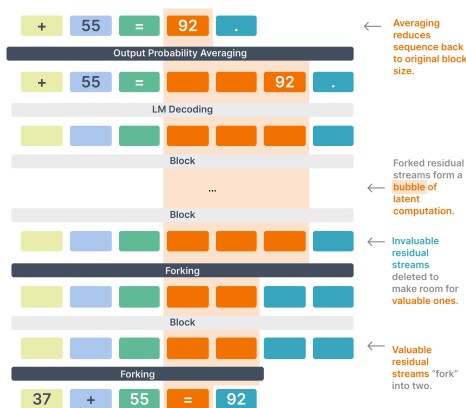

Figure 1: Overview of our method: input tokens fork to form a bubble of latent computation (orange), which is then contracted to produce the final token. Some extraneous tokens may fork (dark blue), but then be pruned.

1. We introduce the first-known architecture to enable the unsupervised dynamic allocation of latent parallel computation, trainable as a regular decoder LM without any additional signal beyond language modeling loss.

2. We demonstrate that our approach performs better in validation perplexity as well in zero-shot evals of LAMBADA and HellaSwag against two competitive baselines—a regular *parameter matched* transformer, as well as a non-adaptive *computation matched* approach where the input residual is copied multiple times as filler tokens for additional computation before decoding. We additionally perform competitively against BLiMP and PIQA.

3. We further show that our method correctly allocates computation at *interpretable* regions of extra computation. In particular, our method allocates more computation at regions of higher uncertainty (i.e., posterior entropy).

We release pretrained adaptive compute LMs and a PyTorch implementation for the community.[1]

## 2 METHODS

### 2.1 OVERALL ARCHITECTURE

Our architecture is a GPT-2-style decoder-only transformer (Radford et al., 2019), trained using the cross-entropy language modeling objective.

To achieve parallel computation, we want to allocate more residual streams corresponding to tokens that require more computation. To enable this, we propose a special type of transformer operation named "forking", described in section 2.3, which can duplicate or remove some input residual streams for future computation.

The amount of forking is controlled by assigning a "cumulative score" between 0 and 1 to each residual stream. Each forking operation computes a "keep score" for each residual stream, which is multiplied to the cumulative score to update it, as well as a new "fork score" for the new residual. We describe the computation of these scores in section 2.3.

---

[1]URL will be available upon acceptance

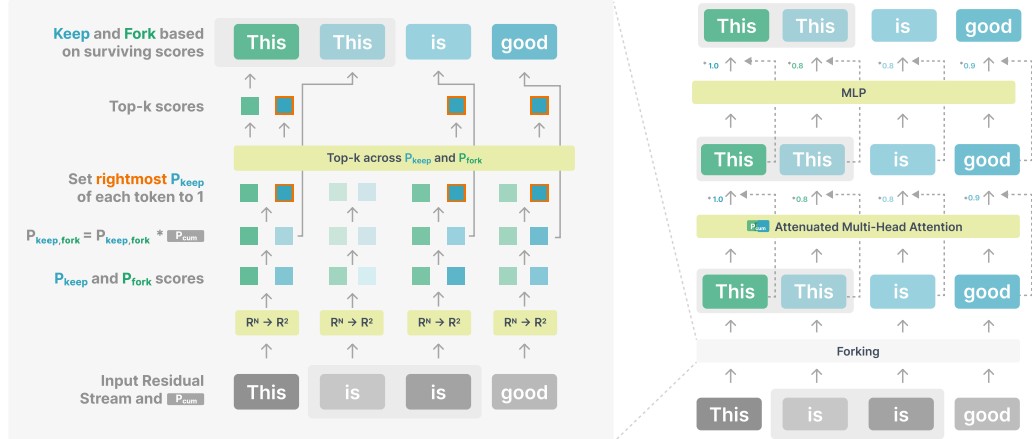

Figure 2: Forking procedure. Token "is" has two forks, one of which will get deleted; the token "this" creates a new fork; we show a score-attenuated transformer block after a forking operation.

This setup reduces the dynamic computation task to determining which residual streams to keep or delete based on the value of cumulative scores: we take the top-k of the scores and perform their corresponding (i.e., keep / fork) actions. As long as the "useful" tokens receive the highest scores, the extra computation should help the performance of the model. To train the model to use the scores correctly, attention and residual updates are attenuated by the cumulative scores (section 2.4). That is, the tokens that the model needs to attend to and update the most become implicitly the highest-scoring tokens to be duplicated.

Additionally, we take special care about the RoPE position embeddings: we apply a "partial rotation" to the forked tokens proportional to the number of forks: the more forks a token has, the "closer together" each of their forks are. This design is described in detail in appendix D.

## 2.2 NOTATION

We will use $x_j^{(k)} \in \mathbb{R}^{d_{\text{model}}}$ to denote the $j^{\text{th}}$ residual stream at the $k^{\text{th}}$ layer. To emphasize that a particular token is the $j^{\text{th}}$ fork of token $i$, we will write $x_{i,j}^{(k)}$. We fork tokens to the left of the original input token. Thus, the original token is always $x_{i,0}^{(0)}$. A sequence of $q$ forks and original token can be written as $\left[ x_{i,q}^{(k)} \ldots x_{i,0}^{(k)} \right]$.

Lastly, we use $L$ to denote the input sequence length (i.e., "input block size", the embedded input to the first block is $x_{1,0}^{(0)} \ldots x_{l,0}^{(0)}$), $N$ to denote the block size at the input to each layer (i.e., before the first layer, $N = L$). We omit the layer index for $N$ to avoid clutter. Additionally, we take a parameter $\kappa$ for the maximum block size. This means that the maximum number of forks each layer is $\kappa - N$.

## 2.3 FORKING

Residual stream insertion and deletion are performed in special forking layers inserted between our score-attenuated transformer blocks, described in section 2.4. Each "forking" layer $k$ parametrized by $\theta$ carries a new "forking decision" function $f_\theta^{(k)} : \mathbb{R}^{d_{\text{model}}} \longrightarrow \mathbb{R}^2$. We apply this new function on each member of the residual stream in order to produce the fork and keep scores, which we then bottleneck using a top-k judgment in order to produce the forked output.

**Scoring.** For each residual, $x_i^{(k-1)}$ (note that the notation here is irrespective of forks or the original token, a distinction which we make later), we first apply the forking decision function along with a sigmoid activation $\sigma$ to obtain a fork and keep scores:

$$\sigma \left( f_\theta^{(k)} \left( x_i^{(k-1)} \right) \right) = \left[ p_{\text{fork},i}^{(k)}, p_{\text{keep},i}^{(k)} \right]. \tag{1}$$

We then update the fork and keep scores inductively based on a "cumulative score" ($p_{\text{cum}}$) propagated from previous layers:

$$\hat{p}_{\text{fork},i}^{(k)} = p_{\text{cum},i}^{(k-1)} \cdot p_{\text{fork},i}^{(k)} \tag{2}$$

$$\hat{p}_{\text{keep},i}^{(k)} = p_{\text{cum},i}^{(k-1)} \cdot p_{\text{keep},i}^{(k)} \tag{3}$$

We initialize the cumulative scores for each input token at the first layer as $p_{\text{cum},(i,0)}^{(0)} = 1$. A subset of these $\hat{p}_{\text{keep}}', \hat{p}_{\text{fork}}$ scores is used as $p_{\text{cum}}^{(k+1)}$, after deciding which ones to keep, as described later.

Note that, in practice, all scores (keep, fork, cumulative) are implemented in log-space for stability instead of being in probability space as shown here.

**Forking Judgments.**   To make sure we have a source token from which to predict each next token, we must ensure that at least one instance is kept throughout the whole model. To do so, we first define a modified keep score that is forced to be 1 (the maximum) for the original, rightmost tokens:

$$\hat{p}_{\text{keep},(k,j)}' = \begin{cases} 1 \text{ if } j = 0 \\ \hat{p}_{\text{keep},(k,j)} \text{ otherwise} \end{cases} \tag{4}$$

Given a set of scores for a layer $k$, we create a list $P = \left[ \hat{p}_{\text{fork},0}^{(k)}, \hat{p}_{\text{keep},0}^{'(k)} \cdots \hat{p}_{\text{fork},n}^{(k)}, \hat{p}_{\text{keep},n}^{'(k)} \right]$, we compute a top-k to downsample this list to obtain $P_\kappa$ where $|P_\kappa| = \kappa$. Using this list, we assemble the new residual stream set $X^{(k)}$ by the following two rules:

$$x_j^{(k)} \in X^{(k)} \text{ if } \hat{p}_{\text{keep},j}' \in P_\kappa \tag{5}$$

$$x_{j_{\text{fork}}}^{(k)} \in X^{(k)} \text{ if } \hat{p}_{\text{fork},j} \in P_\kappa \tag{6}$$

In order to differentiate the forks from their sources, a per-layer learned fork embedding $v_\theta^{'(k)} \in \mathbb{R}^{d_{\text{model}}}$ is added to their parent at initialization: $x_{j_{\text{fork}}}^{(k)} = x_j^{(k)} + v_\theta^{'(k)}$. We arrange the output tokens such that if a new forked residual is created, it is placed to the *left* of its parent.

We define the new cumulative scores $p_{\text{cum}}^{(k)}$ as $\hat{p}_{\text{fork},j}$ for newly forked residuals, and $\hat{p}_{\text{keep},j}$ for kept residuals (note that this is score for which the rightmost token does not have forced-maximum score of 1, allowing the model to ignore the rightmost token if desired.)

## 2.4   RESIDUAL UPDATE ATTENUATION

To learn useful scores, in all blocks, both residual writes and attention computation are modulated by the cumulative scores. Intuitively, this prevents the model from relying on tokens that are about to be deleted due to their low scores.

Specifically, we stack the cumulative scores to a vector $P^{(k)} \in \mathbb{R}^\kappa$:

$$P^{(k)} = \left[ p_{\text{cum},1}^{(k)}, \ldots, p_{\text{cum},\kappa}^{(k)} \right] \tag{7}$$

and use it to modulate both the attention computation and residual updates. We define the attenuated attention operation as:

$$\text{Attn}\left( Q^{(k)}, K^{(k)}, V^{(k)} \right) = \text{softmax}\left( \frac{Q^{(k)} K^{(k)\top} + \mathbb{1} \log\left( P^{(k)} \right)^\top}{\sqrt{d_{\text{model}}}} \right) \left( V^{(k)} \odot P^{(k)} \right) \tag{8}$$

where $\odot$ is the element-wise multiplication. We modify the transformer block (Vaswani et al., 2017) to attentuate the residual whites by $P^{(k)}$ as follows:

$$X^{(k)'} = \text{Attn}\left( f_Q\left( \text{LN}\left( X^{(k)} \right) \right), f_K\left( \text{LN}\left( X^{(k)} \right) \right), f_V\left( \text{LN}\left( X^{(k)} \right) \right) \right) \odot P^{(k)} \mathbb{1}^\top + X^{(k)} \tag{9}$$

$$X^{(k+1)} = \text{MLP}\left( \text{LN}\left( X^{(k)'} \right) \right) \odot P^{(k)} \mathbb{1}^\top + X^{(k)'} \tag{10}$$

for $X^{(k)}$ being the cocatenated list of residual streams in the input of the layer, LN being layernorm, and $f_{Q,K,V}$ being the attention projections. If forking occurred prior to this layer, $X^{(k)}$ is as defined in eqs. 5 and eq. (6) , *after* forking takes place.

## 2.5 OUTPUT AVERAGING

After all transformer layers, we obtain a residual stream set where an input token might be represented by multiple residual streams. To compute a single output distribution for these distributions, we decode each of the residual streams and mix the resulting probability distributions using the cumulative scores. For $\text{Dec}_\theta : \mathbb{R}^d_{\text{model}} \longrightarrow |V|$ being the vocabulary output projection, and $f$ being the last layer of the network, we have:

$$x_i^{(k)} = \frac{1}{\sum_j p_{\text{cum},(i,j)}^{(f)}} \sum_j p_{\text{cum},(i,j)}^{(f)} \text{softmax}\left(\text{Dec}_\theta\left(x_{i,j}^{(k)}\right)\right). \tag{11}$$

We compute this weighted average using the log-sum-exp trick (Blanchard et al., 2021) for stability.

## 2.6 SCORING AND SAMPLING

Because of the possibility of varying $\kappa$ at inference time, there are two main ways inference can be performed in our model. Naively, we can set the inference budget $\kappa_{\text{inference}}$ to be the same as in training time $\kappa_{\text{train}}$, two or four times the block size atr training. We call this **fixed forking**. Alternatively, we can set the inference budget to be the same *ratio* as the training budget. $\kappa_{\text{inference}}$ is set to a value that maintains its same ratio to block size as during training; that, if $\kappa_{\text{train}} = 2l_{\text{train}}$, then $\kappa_{\text{sample}} = 2l_{\text{sample}}$. We call this **dynamic forking**, and discuss this method further in appendix E.1.

**Scoring**    To obtain a probability judgment from our model of a sequence, we provide the entire sequence as input to our model and obtain the posterior probabilities our model assigns to each token of our sequence. For all of our results in table 1, we use dynamic forking.

**Sampling**    We perform autoregression with both fixed and dynamic forking, and discuss the trade-offs of both, in section 5.1. Note that dynamic forking is especially pertinent here because initial sequneces for autoregression is small.

# 3 EXPERIMENTAL SETUP

## 3.1 PARAMETER SELECTION AND TRAINING

Because our architecture takes token embeddings as input and produces token probabilities, it trains exactly like a regular decoder-only language model. As mentioned above, this means that the loss function can be standard language-modeling cross-entropy loss. Optimization is performed by the AdamW optimizer (Loshchilov & Hutter, 2017). Further optimization and architecture details can be found in appendix A.

We insert the first forking layer after a few regular transformer blocks to ensure that the forking score judgments see a broader context window. This is important in order to judge a token's relative importance compared to the others. For all models in section 4, we train models at various scales with token forking placed prior to layers 3, 7, and 11. This means that for models with more layers, the majority of the latter half of transformer will contain no forking. We discuss this choice in appendix B.

## 3.2 PRETRAINING DATASETS

We pretrain our approach on two datasets: `OpenWebText` (Gokaslan et al., 2019), a standard web-text pretraining corpus, as well as `peS2o` (Soldaini & Lo, 2023), a collection of academic papers sourced from the Semantic Scholar Open Research corpus (Lo et al., 2020). Pretraining is conducted for 2.5 billion tokens (75,000 steps). We sample batches at random positions throughout each dataset without additional masking.

## 3.3 BASELINES

**Regular Transformer.**    We first compare against a GPT-2-like (Radford et al., 2019) transformer with RoPE (Su et al., 2024). Our model is based on nanoGPT[2]. We make no changes other than removing the learned position embeddings and including rotational ones in the attention pass.

---

[2]https://github.com/karpathy/nanoGPT

**Duplicated Filler Tokens.** Though a regular transformer is a parameter-matched baseline, our approach will necessarily utilize more computation due to the expanded latent block size (i.e., after forking, the block-size is longer). A naive model of parallel computation that would allow us to slightly exceed the computation of our approach is by simply copying the input residual multiple times before running the transformer, and then taking the rightmost residual for decoding.

### 3.4 PRETRAINING EVALUATIONS

After pretraining, we conduct a variety of zero-shot evaluations on our models and baselines to examine their quality. They include the model's measured perplexity on a holdout validation set, LAMBADA (Paperno et al., 2016) for context extraction, HellaSwag (Zellers et al., 2019) for common sense reasoning, BLiMP (Warstadt et al., 2020) for syntax understanding, and PIQA (Bisk et al., 2020) for embodied physical inference.

For each zero-shot downstream task, we use the dynamic budget as described in appendix E.1. We describe in detail the implementation of the zero-shot evaluations, including what to measure, in appendix E. Across all evaluations, we use the trained models as-is without additional fine-tuning.

## 4 RESULTS

| Dataset | Size | Approach | Perplexity ($\downarrow$) | LAMBADA ($\uparrow$) | HellaSwag ($\uparrow$) | BLiMP ($\uparrow$) | PIQA ($\uparrow$) |
|---------|------|----------|------------|----------|-----------|-------|------|
| OpenWebText | 772M | Baseline | 21.22 | 23.9 | 30.6 | 79.6 | **62.3** |
| | | Copy-3 | 21.20 | 22.8 | 29.0 | 81.2 | 60.4 |
| | | Copy-5 | 20.90 | 19.9 | 29.1 | 80.9 | 60.2 |
| | | Ours ($\kappa = 2L$) | 20.19 | 27.9 | 31.1 | 80.4 | 62.0 |
| | | Ours ($\kappa = 4L$) | **19.74** | **29.4** | **32.25** | **81.6** | 61.9 |
| | 319M | Baseline | 21.56 | 22.1 | 28.7 | 79.0 | 60.5 |
| | | Copy-3 | 21.51 | 21.9 | 28.6 | **80.5** | 60.1 |
| | | Copy-5 | 21.28 | 21.1 | 28.4 | 79.6 | 60.5 |
| | | Ours ($\kappa = 2L$) | 20.55 | 22.9 | **29.3** | 78.3 | **60.9** |
| | | Ours ($\kappa = 4L$) | **20.23** | 23.2 | 29.0 | 78.8 | 60.1 |
| | 150M | Baseline | 24.51 | 18.2 | 26.9 | 76.7 | 57.9 |
| | | Copy-3 | 24.44 | 17.6 | 27.1 | **79.3** | 58.9 |
| | | Copy-5 | 24.40 | 18.9 | 26.9 | 78.8 | 59.4 |
| | | Ours ($\kappa = 2L$) | 23.78 | 21.1 | 27.3 | 77.5 | 59.0 |
| | | Ours ($\kappa = 2L$) | **23.19** | 25.5 | **27.7** | 78.1 | **60.6** |
| peS2o | 772M | Baseline | 14.64 | 9.9 | 27.3 | 69.8 | 55.4 |
| | | Copy-3 | 14.37 | 9.5 | 27.2 | **73.3** | 55.3 |
| | | Copy-5 | 14.50 | 10.3 | 27.3 | 71.6 | 54.5 |
| | | Ours ($\kappa = 2L$) | 13.98 | 10.5 | 27.4 | 68.4 | **56.3** |
| | | Ours ($\kappa = 4L$) | **13.77** | **12.9** | **27.6** | 67.4 | 54.6 |
| | 319M | Baseline | 16.61 | 9.3 | 26.4 | 68.4 | **55.3** |
| | | Copy-3 | 16.41 | 9.4 | **27.2** | **71.8** | 54.7 |
| | | Copy-5 | 16.16 | 8.5 | 26.6 | 70.1 | 55.1 |
| | | Ours ($\kappa = 2L$) | 15.84 | 10.5 | 26.5 | 67.0 | 53.8 |
| | | Ours ($\kappa = 4L$) | **15.61** | 12.3 | **27.2** | 68.6 | 53.6 |
| | 150M | Baseline | 17.10 | 8.1 | 26.4 | 68.6 | 54.5 |
| | | Copy-3 | 16.95 | 7.1 | 26.3 | **69.6** | 54.1 |
| | | Copy-5 | 16.90 | 7.2 | 26.0 | 69.3 | 54.0 |
| | | Ours ($\kappa = 2L$) | 16.90 | 5.0 | 26.2 | 66.6 | **55.1** |
| | | Ours ($\kappa = 4L$) | **16.42** | 10.3 | **26.9** | 67.9 | **55.1** |

Table 1: Zero-shot evaluation results across all model scales after pretraining on 2.5 billion tokens. Each setting is parameter-matched; baseline is a standard GPT-2-like model; copy-3 and copy-5 are models where the input residuals are copied multiple times and can attend to each other; ours is the **thoughtbubbles** transformer, with forking budget set to 2 ($\kappa = 2L$) and 4 ($\kappa = 4L$) times the input block size. The latter of which is roughly FLOPs-matched against copy-5 baseline.

**Our approach performs the best against all baselines in validation perplexity, even exceeding models of bigger scale.** Across both parameter and computation matched settings, we find that our model scores the lowest perplexity across all evaluations. Figure 3 highlights the scalability of our approach: surprisingly, our approach at a 319M parameter scale has lower perplexity on OpenWebText than the baseline approach at the 772M scale.

**Across most zero-shot evaluations, our approach outperforms baselines.** For all LAMBADA and HellaSwag evaluations, we find that our approach outperforms both the parameter-matched baselines as well as the computation-matched baselines. However, we note that for BLiMP (syntax understanding) our model only outperforms the parameter-matched, but not computation-matched baselines—indicating that pruned dynamic parallel computation may not be as helpful for syntax matches. Finally, our model performs similarly to the baseline for embodied reasoning. We attribute this degraded performance to the fact that a short (2.5BT) training may not capture enough information for the embodied NLI to be effectively measured.

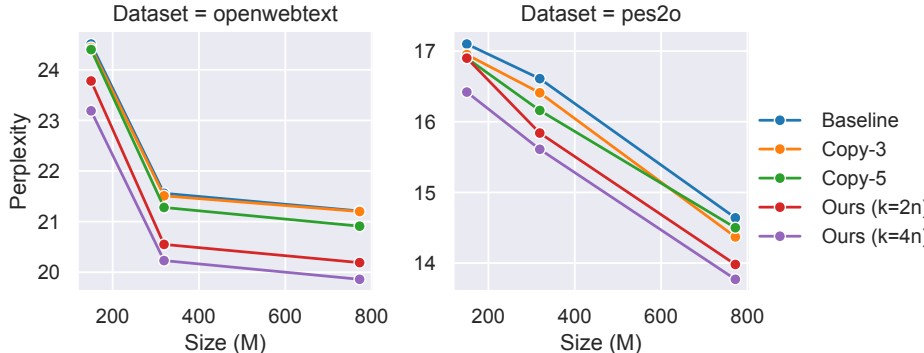

Figure 3: Dev-set perplexity of our approach and various baselines as a function of model scale on both `OpenWebText` and `peS2o` datasets. Across all scales, note that our method outperforms all baselines, including both computation and parameter-matched ones. Lower is better.

## 5 ANALYSIS

**Forks meaningfully influence the value of the parent token.** In fig. 4, we see the rightmost ("og") token attends to its children with attention scores more than an order of magnitude higher than other tokens—second only to attention of those tokens to themselves. This result indicate that the forking tokens play a large role in the computation of the residual update for the rightmost token than most other tokens, indicating their utility in computing the final output.

**Our model dynamically allocates more computation at regions of higher uncertainty without explicit supervision ...** Despite no explicit interventions or regularization, our method learned to allocate more computation at areas of greater uncertainty. We see in fig. 5 that our method

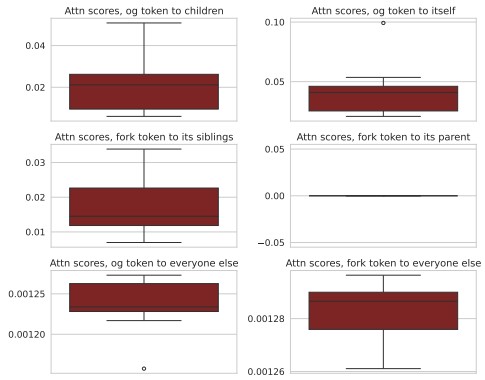

Figure 4: Analysis of attention allocation between the main (rightmost, "og") token and its child forks on our approach trained on `openwebtext`. Note that since we place child token embeddings to the to the left of the main token, forked children cannot attend to its parent.

allocates more computation tokens with high output distribution entropy; this is true both for the entropy measured from the forking model as well as an independently trained, parameter matched decoder LM that does not fork. This is in-line with recent literature (Wang et al., 2025) that highlights the informativeness of high entropy tokens.

**... but will reduce computation at areas of greatest uncertainty.** Despite the previous point, however, we note that our model allocates relatively less budget at tokens of the highest uncertainty, forming a concave parabolic relationship between entropy and computation allocation. We hypothesize that this is due to the relatively higher utility of further computation at areas of moderate (but not

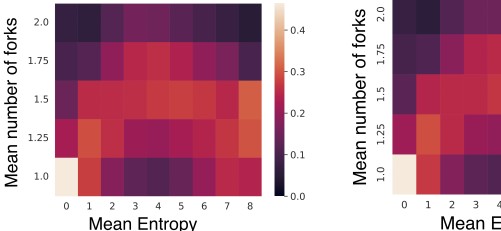

Figure 5: Normalized number of forks in the final layer across a window of 4 tokens as a function of the mean entropy of those 4 tokens on `OpenWebText`. Left: entropy as measured by the forking transformer; right: entropy as measured by a baseline decoder LM.

low) uncertainty: for instance, while choosing between a few options; conversely, areas of highest uncertainty are often caused by the edges of clauses or coreferences, where additional computation will not help resolve the uncertainty.

## 5.1 AUTOREGRESSION

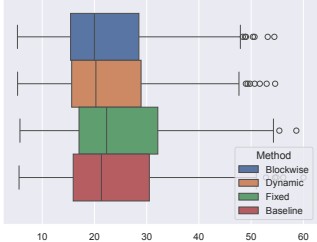

| Approach | Perplexity |
|---|---|
| Ours (Blockwise) | **20.97** |
| Ours (Fixed Budget) | 23.10 |
| Ours (Dynamic Budget) | 21.18 |
| Baseline | 22.15 |

Figure 6: Perplexity distribution and mean perplexity of our 772M ($\kappa = 2L$) model over smaller subset of `OpenWebText` dev set between blockwise forward versus autoregression. Left: naive autoregression; right: autoregression with forking budget proportional to input size. Lower is better.

As seen in fig. 6, implementing autoregression naively with a fixed block size irrespective of the input sequence length results in a distribution shift between blockwise forward pass and autoregression—since the maximum allowed number of forks is much higher if input sequence length is smaller while the total budget remains the same.

However, if we apply the forking budget scaling mitigation described in appendix E.1, we find that our model performs roughly equivalently to the blockwise forward pass, and retains our approach's performance gains above baseline. This result indicates that, while our result can adapt to different inference-time input sizes, care must be taken to scale the adaptive computation budget accordingly.

## 6 RELATED WORK

**Chain-of-Thought Approaches**  Chain-of-thought (Wei et al., 2023) is a simple form of adaptive computation which uses natural-language-based autoregression with additional tokens to achieve thinking. Variants of this approach include simply supervising the output chain (Zhang & Ding, 2024), to replacing them with continuous traces (Hao et al., 2024) or controlled non-adaptive filler tokens (Pfau et al., 2024). Unlike chain-of-thought, our method performs adaptive computation not with recurrence but parallel computation, improving efficiency as well as being able to be trained without additional supervision.

**Adaptive Computation**  Methods vary to force a dynamic amount of computation from a neural model based on the problem. The oldest approaches involves explicitly forcing recurrence (Graves, 2016), while modern LMs yield performance improvements through forcing very simple interventions to existing chain of thoughts (Muennighoff et al., 2025), skipping or adding recurrent compute across layers without adding additional streams of computation (Dehghani et al., 2019; Csordás

et al., 2024; Chen et al., 2024; Murty et al., 2023; Raposo et al., 2024; Kallini et al., 2024), or—most similar to our approach—by adding additional residual streams when computation is needed (Herel & Mikolov, 2024; Goyal et al., 2024; Sun et al., 2025). Unlike prior art, our method removes the need to insert latent tokens explicitly during training or inference, but still gives the ability to gain additional streams of computation through latent residual streams.

**Analysis of Latent Computation**   There's a large and robust literature on the complexity-theoretic power of transformers. Results have shown the limited expressive power of standard transformer computation (Merrill & Sabharwal, 2023), and the additional power that chain-of-thought or even padding tokens add to the computation (Merrill & Sabharwal, 2025; London & Kanade, 2025). Work has also shown the limitations given by single special-token thinking approaches that are not input adaptive (Vennam et al., 2024). Prior work have also shown through techniques in intepretability that even simple chain of thought computation carries implicit intermediate computation similar to depth-bounded recurrence (Brinkmann et al., 2024). We also demonstrate here the power of adaptive latent computation in our work by demonstrating its superior performance even against computation matched baselines; furthermore, we demonstrate that we are indeed performing additional computation in "decisive" high entropy tokens, in line with prior analyses (Wang et al., 2025).

## 7   CONCLUSION

In this work we introduce **thoughtbubbles**, the first adaptive parallel computation architecture that's 1) trainable without additional supervision beyond LM loss 2) allocates computation and memory at interpretable regions of uncertainty and 3) performs better than baseline models in both perplexity and across a suite of zero-shot evals on both parameter-matched and computation-matched settings.

This method unlocks the previously missing input-adaptivity of transformer computation, which allows our model to solve more difficult tasks that require scaling inference-time computation. We demonstrate the efficacy of our method via a suite of zero-shot evaluations on models pretrained on both `OpenWebText` and `peS2o` in both computation and parameter matched settings. Excitingly, our method at a smaller 319M scale outperformed baselines at 772M scale.

Most importantly, our model enables learning latent adaptive computation in a language model already during the pre-training phase. Unlike CoT approaches, it does not rely on being exposed to step-wise instructions during pre-training. We hope that this will unlock a new generation of transformer architectures with more general latent computation, which in turn enables more helpful and capable adaptive compute models.

## 8   LIMITATIONS

**Time-matched evaluations**   Our current approach is implemented in raw PyTorch, requiring no other hardware-level adaptations apart from scatter-max kernels.[3] As such, though our implementation exceeds the performance of a computation-matched approach, its raw wall-clock efficiency is relatively low. Further efforts in implementing hardware-adaptive kernels for operations like forking gather will enable faster computational performance.

**Top-K Gradient Bottleneck**   While forking in earlier layers improves the performance of our models, we note that, interestingly, too much forking results in no further performance improvement at a fixed block size budget (appendix B). We believe this is due to certain tokens with high cumulative scores early on in the model being dropped by hard top-k decisions later in the model, thus resulting in no gradients to update the early large cumulative scores. By implementing training time randomization and noise, this can be mitigated to improve deep forking performance.

**Downstream Reasoning Tasks**   Given our hardware limitations, we cannot experiment with approaches at sufficient scale to enable non-noisy measurements on hard reasoning datasets such as GSM8k (Cobbe et al., 2021), which—without customized training regimes—usually emerges with good performance only around the multi-billion-parameter scale (Liu et al., 2023). In future work, we hope to perform these evaluations with additional resources.

---

[3] https://github.com/rusty1s/pytorch_scatter

## USE OF LARGE LANGUAGE MODELS

We use LLMs for copyediting only. All ideas and content contained herein are our own.

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

## A    EXACT BENCHMARK MODEL ARCHITECTURE

Our benchmark runs involve a variety of model configurations across different scales. All models were trained with a shared optimization configuration, detailed in table 2. Optimization was performed using mix-precision training using `bfloat16`, but with the cumulative forking scores tracked in log space in `float32`; we chose to do this in particular because small numerical imprecision forking judgments may result in large top-k outcome differences.

A vocab size of 50304 to optimize for tensor core efficiency is used, resulting in 47 unused tokens. Forking layers are placed in layers 3, 7, and 11–irrespective of $N_{\text{layers}}$ of the design.

| Hyperparameter | Value |
|---|---|
| Maximum Learning rate | 2.5e-4 |
| Warmup fraction | 0.01 |
| Optimizer | AdamW |
| Weight decay | 0.1 |
| Warmup | 0.01 |
| $\beta_1$ | 0.9 |
| $\beta_2$ | 0.95 |
| Dropout | 0.0 |
| Bias | True |
| Batch size (global) | 64 |
| Steps (global) | 75000 |
| Vocab size | 50304 |
| Block size | 512 |

Table 2: Optimization parameters shared across all scales; note that the actual per-machine batch size differs based on model architecture, the details of which is listed below.

Each model scale has share a common implementation, but contains different topology configurations which increases its parameter count; these configurations are enumerated in table 3.

| Size | Approach | $N_{\text{layers}}$ | $N_{\text{heads}}$ | $d_{\text{model}}$ | Batch | Accumulation | Expanded Size |
|---|---|---|---|---|---|---|---|
| 150M | Baseline | 16 | 12 | 768 | 8 | 8 | 512 |
| 150M | Copy-3 | 16 | 12 | 768 | 8 | 8 | 1536 |
| 150M | Copy-5 | 16 | 12 | 768 | 8 | 8 | 2560 |
| 150M | Ours ($\kappa = 2L$) | 16 | 12 | 768 | 8 | 8 | 1024 |
| 150M | Ours ($\kappa = 4L$) | 16 | 12 | 768 | 8 | 8 | 2048 |
| 319M | Baseline | 24 | 16 | 1024 | 4 | 16 | 512 |
| 319M | Copy-3 | 24 | 16 | 1024 | 4 | 16 | 1536 |
| 319M | Copy-5 | 24 | 16 | 1024 | 4 | 16 | 2560 |
| 319M | Ours ($\kappa = 2L$) | 24 | 16 | 1024 | 4 | 16 | 1024 |
| 319M | Ours ($\kappa = 4L$) | 24 | 16 | 1024 | 4 | 16 | 2048 |
| 772M | Baseline | 36 | 20 | 1280 | 2 | 32 | 512 |
| 772M | Copy-3 | 36 | 20 | 1280 | 2 | 32 | 1536 |
| 772M | Copy-5 | 36 | 20 | 1280 | 2 | 32 | 2560 |
| 772M | Ours ($\kappa = 2L$) | 36 | 20 | 1280 | 2 | 32 | 1024 |
| 772M | Ours ($\kappa = 4L$) | 36 | 20 | 1280 | 2 | 32 | 2048 |

Table 3: Model topology parameters for each scale

Optimization of each run is conducted on a single NVIDIA H200 GPU. Dataset tokenization uses the pre-trained tokenizer from GPT-2 (Radford et al., 2019). FlashAttention kernels (Dao et al., 2022) are used to train our system, with value vector attenuation occurring before.

## B  OVERFORKING

We perform a minimal ablation to examine if additional layers of forking beyond layers 3, 7, and 13 would help model performance. In particular, we trained our 772M scale model for 25,000 steps with forking at layers 3, 7, and 11 only as well as extended into all layers $4n - 1$ (i.e., 16, 20, ...) thereafter.

| Approach | Perplexity |
|---|---|
| Ours | 29.84 |
| Ours (extended forking) | 28.02 |

Table 4: Performance of an ablation where we performed more forking at later layers, trained on `OpenWebText` for 25,000 steps (roughly 0.8BT). We see that the extended forking approach is slightly worse than forking only in the beginning.

## C  ANALYSIS OF FORKING LOCATIONS

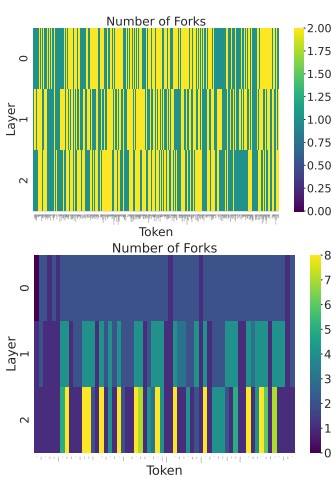

Figure 7: Number of token forks created by the model at each layer for an input sample of `OpenWebText` (top), and number of forking tokens created by the model on a sample of `CLUTTR` (bottom).

We perform here a qualitative analysis of where our model allocates computation. In particular, after training, we plot the number of residuals each input token is forked into after each forking judgment. We run this analysis on both a sample of `OpenWebText` , as well as a synthetic task with known "difficult" computation locations—a relational graph ST-connectivity task named `CLUTTR` (Sinha et al., 2019).

In fig. 7, we qualitatively observe that our model has learned to perform extra computation near intepretable decision boundaries for synthetic tasks. For `CLUTTR`, forking occurs near coreferent entities and at special tokens delineating the beginning of query components or response. In contrast, the result for `OpenWebText` shows that computation on web-text is spread evenly across the sequence—namely, that its not sequence position dependent, as it is for synthetic tasks such as `CLUTTR` due to their structure.

This result, along with the result in fig. 5, indicates that our model can truly dynamically allocate computation to the areas of the biggest greatest computational difficulty and is not relying on a simple heuristic for allocating computation.

## D  POSITION ENCODING

Due to the fact that this architecture introduces multiple possible residuals for every input token, care must be taken to ensure that position embeddings scale by the amount of forking accordingly. In order to do this, we implement a Rotational Position Embedding (RoPE, Su et al. (2024)) variant to offset smaller rotations degrees when there are more forks.

Recall that, typically, RoPE is defined, for $x_k^{(j)}$ being the $j$-th slot of the residual stream of token $k$,

$$\text{RoPE}\left(x_k^{(i)}, x_k^{(j)}, k\right) = \begin{pmatrix} \cos k\theta & -\sin k\theta \\ \sin k\theta & \cos k\theta \end{pmatrix} \begin{pmatrix} x_k^{(i)} \\ x_k^{(j)} \end{pmatrix}. \tag{12}$$

where $\theta$ is the total rotation angle. In our approach, however, we may have $q$ streams representing a particular input token. That is, token $k$ is forked into residual streams $x_{(q-1),k}, \ldots, x_{0,k}$. In order to accommodate tokens of different number of forks, we augment RoPE with *partial* rotations proportional to the number of forks of each token. For the $i,j$-th slot of residual $p$ of token $k$ which has $q$ forks in total, we write:

$$\text{RoPE}\left(x_{p,k}^{(i)}, x_{p,k}^{(j)}, k\right) = \begin{pmatrix} \cos\left((k - \frac{p}{q})\theta\right) & -\sin\left((k - \frac{p}{q})\theta\right) \\ \sin\left((k - \frac{p}{q})\theta\right) & \cos\left((k - \frac{p}{q})\theta\right) \end{pmatrix} \begin{pmatrix} x_k^{(i)} \\ x_k^{(j)} \end{pmatrix}. \tag{13}$$

That is, the more forks a particular token has, the "closer together" in position embeddings each of its forks will be.

## E  DETAILS ON ZERO-SHOT EVALS

**Perplexity**  We first evaluate the perplexity score of each model against the development sets of the respective datasets. This is our primary measure of quality, as it represents our approache's ability general ability to model text effectively.

**LAMBADA**  To explore our model's ability to extract useful information from context, we further evaluate the approach on the Lambada dataset (Paperno et al., 2016), a final-word prediction dataset for the correct answer is heavily dependent on detail revealed in context long before the final word. Given the entire context, we predict only the final word (i.e. space-delineated run of tokens) and compare against "gold"; a task is solved correctly if the final word exactly matches.

**HellaSwag**  To evaluate our model's knowledge and natural-language inference (NLI) capabilities, we perform evaluations against the HellaSwag dataset (Zellers et al., 2019)—a common-sense based NLI dataset. We concatenate each continuation against the premise and evaluate the perplexity of each. A task is solved correctly if the lowest-perplexity sequence is the target sequence.

**BLiMP**  To evaluate our model's syntax understanding, we evaluate our model's performance across all splits of the BLiMP dataset (Warstadt et al., 2020). The dataset contains pairs of lexically similar sequences, but only one of which is syntactically sound. A task is solved correctly if the model assigns lower perplexity to the grammatical sequence.

**PIQA**  Finally, to evaluate our model's knowledge and embodied common sense, we evaluate our model's performance on PIQA—an NLI style dataset for physical reasoning. (Bisk et al., 2020) As with HellaSwag, we concatenate each continuation against the premise and evaluate the perplexity of each. A task is solved correctly if the lowest-perplexity sequence is the target sequence.

### E.1  INFERENCE-TIME BUDGET SCALING

Inference with short sequences on our method yields a distribution shift: if $\kappa$, the maximum block size, is kept the same for any block size, short sequences would be able to fork many more times

than longer ones. This problem is especially prevalent during autoregression, where the initial input sequence is much shorter than the full block size.

To mitigate this, we scale the inference time forking budget $\kappa$, *proportionally* to the full-width block size. Specifically, we compute at training-time ratio $r = \frac{\kappa}{L}$ for training-time block size $L$ and maximum budget $\kappa$; at inference time, for an input of size $L'$, we set a temporary max budget $\kappa' = rL'$ for the top-k operation. This can also be understood as a "rolling top-k" operation that is iteratively updated at each token. This trick enables our method to work autoregressively with minimal performance degradation.

