# OpenReview forum: "Thoughtbubbles: an Unsupervised Method for Parallel Thinking in Latent Space"
_ICLR.cc/2026/Conference — Submitted to ICLR 2026_

### Official Review · Reviewer_Ck1v · 2025-10-28

**Soundness:** 2
**Presentation:** 3
**Contribution:** 2
**Rating:** 4
**Confidence:** 4

**Summary:**

This paper proposes a new framework, Thoughtbubbles, for latent parallel computation. They enable the dynamic allocation strategy to introduce residual streams, but delete or folk them based on judgments. With their folking decision methods and tailored structures, Thoughtbubbles achieve better perplexity and zero-shot accuracy.

**Strengths:**

- The paper is well-written in overall.
- Adaptive and dynamic strategy seems a good motivation for latent thinking approaches.
- Overall method (approach) is well aligned with the intuitions, and potential issues (like RoPE) are also explained throughout the paper.

**Weaknesses:**

- First of all, 2.5B tokens are too small to get valuable insights I believe. At least 20 times larger tokens to model size would be great to support the performance improvement.
- Are all results averaging all residual streams? Do you have any result without output averaging to see if latent thoughts have good effects themselves?
- Since longer sequences will introduce extra computation time, it'd be good to have those kinds of results for scaling effects.
- I also curious about the batching inference latency itself. These dynamics will hurt the performance.
- Could you elaborate 'log-space' in L.171?
- Regarding Eq.8 and 9, are there any ablation study results? It seems like attention outputs are affected by square of P (cumulative scores), a little bit weird to me.
- In Appendix D (Eq.13), the last fork (p = q) will get the same position embeddings as the previous step token, right?
- Missing reference: Mixture-of-Recursions paper seems also be related to dynamic allocation of latent thoughts (vertical manner of recursion can be seen as horizontal manner.) The mechanism of folking or deleting inside seems different though. Nevertheless, it would be good to add discussion with that kinds of literature.

**Questions:**

See above weakness parts.

---

> ### Author Response · Authors · 2025-11-17
>
> > First of all, 2.5B tokens are too small to get valuable insights I believe. At least 20 times larger tokens to model size would be great to support the performance improvement.
>
> While it is always better to evaluate every novel idea on bigger models trained on more tokens, unfortunately an exhaustive experiment across large data scales is not feasible on an academic budget.
>
> Note that we demonstrated improved performance across the 100M–700M scaling ladder, and in particular:
>
> From [1][2], recall that the optimal data scale follows:
>
> $C = 6ND$
>
> Where $C$ is the total flops, $N$ is the number of parameters, and $D$ is the token count.
>
> Now, figure 4 in [1] highlights that the optimal model size for flops count of $10^{19}$ is roughly $700\ \text{million}$ parameters, as in our experiments.
>
> Using the expression above, we obtain that the optimal data scale for this comparison is around 2.3e9 tokens.
>
> We train for 2.5e9 tokens, as follows this expression.
>
> We note that the scaling laws of different architectures would perform differently, yet in our work we perform strongly against isoToken measurements; we further include here the performance of our approach compared to two times the token usage.
>
> [1] Hoffman, et al., “Training Compute-Optimal Large Language Models” (2022)
>
> [2] https://discuss.huggingface.co/t/total-flos-vs-c-6-n-d/65593
>
> > Are all results averaging all residual streams? Do you have any result without output averaging to see if latent thoughts have good effects themselves?
>
> Yes, all results are using cumulative scores as a weighted average over residual streams. We note that keeping rightmost only worsens performance, as shown in the ablation study regarding keeping the rightmost token in our overall response above.
>
> > Since longer sequences will introduce extra computation time, it'd be good to have those kinds of results for scaling effects.
>
> Agreed, and we can run this ablation for camera ready. Note that this is a different form of computation than the form that we present here, as autoregression on longer sequences requires expanding sequential computation, whereas out approach induces additional computation in parallel. Furthermore, we note that our approach performs better against computation exceeded baselines using 2x more data, as detailed above.
>
> > I also curious about the batching inference latency itself. These dynamics will hurt the performance.
>
> As described in section 5.1, we implement inference-time autoregression at the same ratio of maximum as in training time. Yet, during our computation matched experiments (and as discussed in the additional experiments here titled “More Computation Matching Experiments”), we found that our method outperforms approaches at the same scale even when expending additional compute. Thus, we would expect that our method trades off an equivalent ratio of performance to latency at inference time as well.
>
> One way to optimize serving our model is to perform a dynamic batching whereby we batch more sequences in early layers to take advantage of the existing launched kernel. Any method that leverages inference-time adaptive computation will suffer from a degree of strided batching whereby the adaptive compute’s output shape is not predictable ahead of time.
>
> > Could you elaborate 'log-space' in L.171?
>
> Instead of storing and propagating $p_{cum,fork,keep}$, we store and propagate $log(p_{cum,fork,keep})$. As in, when we need to perform multiplication of other values against the scores, we take the scores and exponentiate them before using. Thus to multiply two scores together we simply add them in practice. This helps prevent underflow issues early on in training.
>
> > Attention outputs are affected by square of P (cumulative scores), a little bit weird to me.
>
> Note that the attention provides the main learning signal to our forking scores. We believe the double-scaling is beneficial because the gradient propagation through the attention matrix is weaker (due to the softmax) than through the values directly. On the other hand, if we just down-weight the values, the softmax will still take into account the deleted tokens, potentially taking away useful attention mass from the persistent tokens.
>
> > In Appendix D (Eq.13), the last fork (p = q) will get the same position embeddings as the previous step token, right?
>
> No. Recall forks are written from left to right: $x_{(q-1)} \;\dots\; x_0$. Hence the last fork would be $\sin/\cos(k - 0/q) = \sin/\cos(k)$. For the first fork, it would be written as $\frac{q-1}{q}$, which is $\frac{1}{q}$ behind $\sin/\cos(k - 1)$, which is the previous token.
>
> > Missing reference: Mixture-of-Recursions paper seems also be related
>
> Thank you for the reference, we will add the reference to our article.

---

> > ### Comment · Reviewer_Ck1v · 2025-11-25
> >
> > Thanks for the detailed responses.
> >
> > C = 6ND is just how to approximately calculate the FLOPs of models. Since backward has twice FLOPs than forward and there are add and multiply operations, it becomes 6 times N D (most parameters are linear).
> > That's why if you choose your FLOPs budget as 10^19, then D will be around 2.5B just because you are using 700M model.
> > See Table 3 in [1]. 20 times larger than the model would be optimal. That's why I asked at least 20 times larger training tokens, although I truly understand academic budget is not enough even for that.
> >
> > And in compute-bound scenarios, will this method be more expensive? Parallel computation will also be the bottleneck in compute-bound.
> > Performance seems to be well-scaled, but not sure about the efficiency. Even though parallel decoding can be good during memory-bound, this can increase the latency much more for longer contexts.
> >
> > I'll keep my score accordingly.

---

### Official Review · Reviewer_7Vnw · 2025-11-02

**Soundness:** 2
**Presentation:** 2
**Contribution:** 1
**Rating:** 2
**Confidence:** 4

**Summary:**

The paper proposes Thoughtbubbles, a transformer architecture that dynamically allocate computational resources according to the learned cumulative scores. The scores are computed for each token across layers; authors use top-k selection to keep high-scoring tokens and weigh attention by these scores. The method is applied to language model pretraining with OpenWebText and peS2o and shows improvements in perplexity and some zero-shot tasks over baselines.

**Strengths:**

* Authors propose a forking mechanism that is conceptually lightweight and also compatible with modern language model pretraining
* The paper is well motivated and well written; despite complex notations it is very readable
* Experiments are conducted over two pretraining sets and results on both perplexity and zero-shot tasks are reported in detail

**Weaknesses:**

* One major concern I have regarding the paper is its claim on "reasoning". The paper repeatedly claims "latent computation" and "parallel thinking in latent space" but the proposed method to its essence is a weighted transformer with token duplication, not a latent variable model with explicitly modeled "reasoning" process. Recent works demonstrate how explicit reasoning can be included in the pretraining process, for example [1][2]. I recommend the authors discuss these works, and either reframe the contribution as "adaptive attention allocation" or provide rigorous justification for why duplicated, masked token embeddings constitute "latent" computation.

* Regarding experiment results -- the paper claims to enable reasoning but show little improvements on reasoning tasks. For example, PIQA is focused on physical reasoning, but the proposed method performs similar to baselines; BLiMP only beats parameter-matched baselines. For math reasoning, authors mention that GSM8k is "too noisy at this scale", but reasoning LMs of similar scale like [2] has demonstrated reasoning capabilities using the dataset. The model showed increase in performance mostly on tasks like Lambada, which are essentially context-retrieval-based test sets, and thus can only show mechanism improves attention bandwidth, but not helping with reasoning.

* The experiments also lack baselines. The primary baseline is a "Copy-N" model that naively duplicates inputs. Authors should also incorporate results from related work (some already discussed in the paper) like [3][4].

* The paper introduces a "partial rotation" scheme for forked tokens. However this seems more like an ad-hoc design and there is no theoretical justification for fractional position offsets.

* The forking mechanism relies on hard top-k selection to determine which token streams survive at each layer. While this enables explicit pruning, I wonder if it creates a potential gradient flow problem, such that tokens whose scores fall just below the top-k threshold receive no gradient signal about how to improve.

[1] Hao, Shibo, Sainbayar Sukhbaatar, DiJia Su, Xian Li, Zhiting Hu, Jason Weston, and Yuandong Tian. "Training large language models to reason in a continuous latent space." arXiv preprint arXiv:2412.06769 (2024).

[2] Kong, Deqian, Minglu Zhao, Dehong Xu, Bo Pang, Shu Wang, Edouardo Honig, Zhangzhang Si et al. "Latent Thought Models with Variational Bayes Inference-Time Computation." arXiv preprint arXiv:2502.01567 (2025).

[3] Raposo, David, Sam Ritter, Blake Richards, Timothy Lillicrap, Peter Conway Humphreys, and Adam Santoro. "Mixture-of-depths: Dynamically allocating compute in transformer-based language models." arXiv preprint arXiv:2404.02258 (2024).

[4] Dehghani, Mostafa, Stephan Gouws, Oriol Vinyals, Jakob Uszkoreit, and Łukasz Kaiser. "Universal transformers." arXiv preprint arXiv:1807.03819 (2018).

**Questions:**

* Can the authors report results on GSM8K? I am curious about how it scales as the model size increases
* Figure 5 claims the model allocates more computation to high-entropy tokens as evidence for intelligent computation allocation. I wonder if you remove forks from high-entropy tokens and add them to low-entropy tokens, does performance drop more than the reverse?
* See additional questions in the weakness section

---

> ### Author Response · Authors · 2025-11-17
>
> > One major concern I have regarding the paper is its claim on "reasoning". The paper repeatedly claims "latent computation" and "parallel thinking in latent space" but the proposed method to its essence is a weighted transformer with token duplication, not a latent variable model with explicitly modeled "reasoning" process.
>
> Thank you for the references. We have a limited discussion of these works in our article, in particular [1] as reference above, because it's a different form of computation—i.e. sequential stepwise computation—than the form we are interested in, which is parallel computation. Our approach has a significant efficiency advantage because parallel adaptive computation can be parallelized during each forward pass instead of needing the previous token to perform autoregression. However, we will update the language in our article, in particular with respect to “reasoning”, as detailed above.
>
> > Regarding experiment results -- the paper claims to enable reasoning but show little improvements on reasoning tasks.
>
> While PIQA performance is mixed, note that in table 1 we observe increased performance over all baselines as model size increases. Therefore, we believe that at scale our performance will become strengthened. We will train a 1B+ parameter model for camera ready, and we expect similar scale improvements. For GSM8k, to the best of our knowledge there is currently no publicly known method that enables training a model from scratch on fully academic budget and demonstrating non-noisy performance on evaluations like GSM8k. [2] uses a generous pass@5 metric and allocated much more token budget, yet a GPT-2 scale model still achieves <8% accuracy on the dataset.
>
> > The experiments also lack baselines. The primary baseline is a "Copy-N" model that naively duplicates inputs. Authors should also incorporate results from related work (some already discussed in the paper) like [3][4].
>
> We perform further baselines for our experiment in the section above titled “more computation matched experiments” where we show the performance of our model exceeds that of a regular transformer trained on double the amount of data.
>
> > The paper introduces a "partial rotation" scheme for forked tokens. However this seems more like an ad-hoc design and there is no theoretical justification for fractional position offsets.
>
> Partial angle rotations provide a natural extension of the intuition of rope, where pairwise angle distances represent distances. An fractional add to the rotation RoPE angle has been explored previously in literature that show its ability to aid length extrapolation. [1]
>
> > The forking mechanism relies on hard top-k selection to determine which token streams survive at each layer. While this enables explicit pruning, I wonder if it creates a potential gradient flow problem, such that tokens whose scores fall just below the top-k threshold receive no gradient signal about how to improve.
>
> We find that, while forking too much may create a gradient flow problem by saturating the top-k bound, a conservative amount of forking in earlier layers improves performance. We see this effect in Appendix B, whereby forking too much in later layers results in reduced performance. For future work, one approach that can be explored for gradient  flow involves merging deleted tokens back into residual stream, leading to no bottleneck for deleted tokens.
>
> > Figure 5 claims the model allocates more computation to high-entropy tokens as evidence for intelligent computation allocation. I wonder if you remove forks from high-entropy tokens and add them to low-entropy tokens, does performance drop more than the reverse?
>
> We find that naively doing this zero-shot results in a significant (i.e. val loss averages to 9.2) drop in performance, though this is not meaningful as the model is not used to being manipulated for token allocation zero-shot. Recent work have show that searching on these high-entropy tokens are indeed decisive for computation in harder task [2]. We would love clarity with respect to what the desired measurement is here.
>
> [1] Sun, et al., 2025 “A Length-Extrapolatable Transformer."
>
> [2] Wang, et al., 2025 "High-Entropy Minority Tokens Drive Effective Reinforcement Learning for LLM Reasoning"

---

### Official Review · Reviewer_nkbZ · 2025-11-06

**Soundness:** 2
**Presentation:** 2
**Contribution:** 2
**Rating:** 4
**Confidence:** 4

**Summary:**

The paper proposes Thoughtbubbles, a decoder-only transformer that adaptively allocates parallel compute by forking or deleting residual streams per token. A learned cumulative score governs when to fork or prune; low-score streams are attenuated (masked from attention and updates), while high-score streams are duplicated. After the stack, per-token distributions from all surviving streams are score-weighted averaged to produce the final logits. The system is trained only with standard LM cross-entropy (no extra supervision) and shows lower perplexity than (i) a parameter-matched GPT-2-style baseline and (ii) a non-adaptive, “Copy-N” computation-matched baseline on OpenWebText and peS2o; it also reports gains on LAMBADA and HellaSwag.

**Strengths:**

1. Novel Architecture: The core idea of an unsupervised, adaptive, and parallel compute mechanism is conceptually clean, novel, and a valuable research direction.

2. Perplexity Gains: The method demonstrates clear and consistent empirical wins on perplexity across all tested scales (150M–772M) and corpora when compared to the provided baselines (Table 1, Fig. 3).

3. Insightful Analysis: The analysis in Figure 5, showing that the model allocates more compute to higher-entropy regions, provides compelling evidence that the adaptive mechanism is learning non-trivial, interpretable behavior.

**Weaknesses:**

1) Reasoning Claims Are Not Supported by Appropriate Evaluations
The paper's most significant weakness is the disconnect between its framing and its results. The work is motivated as an advance toward "thinking" beyond Chain-of-Thought (CoT), yet it is not evaluated on any of the standard multi-step reasoning benchmarks (e.g., GSM8K, MATH, Big-Bench Hard) where such capabilities are measured. The authors acknowledge this as a limitation, but its omission is critical. Without this data, the central claim—that the "bubbles" constitute meaningful "thinking"—is unsubstantiated.

2) Inadequate Computational Baselines and Cost Analysis
The paper's claims of outperforming "computation-matched" baselines are not robust, for three key reasons:

(1) Weak FLOPs-Matched Baseline: The "Copy-N" baseline is a poor and non-competitive comparator. It measures the effect of applying a small model to more data, not the performance of a different model with an equivalent compute budget. A far more competitive and necessary baseline is a standard, non-adaptive GPT-2-style model that is scaled up (e.g., made wider or deeper) to have the same training and/or inference FLOPs as the "Ours (k=4L)" model. It is very plausible that a standard, larger transformer would outperform this complex, adaptive one.

(2) Missing Wall-Clock Comparison: The paper omits a practical, wall-clock time comparison. The authors admit to "low raw wall-clock efficiency" due to the overhead of forking, gating, and averaging. A fair comparison requires measuring throughput/latency against all baselines (including a standard larger-model baseline) on a time-matched budget.

(3) Missing Adaptive Baselines: The paper fails to compare against other, more established adaptive compute methods, such as layer-skipping, early-exits, or mixture-of-depths. Without these, it's impossible to know if this parallel-forking approach is superior to adaptive-depth.

3) Key Ablations Are Missing
The paper does not provide sufficient ablation studies to isolate which components of the method are responsible for the gains. For example, what is the impact of the attenuation mechanism (masking updates/attention by score) versus the forking alone? How sensitive is the model to the placement of forking layers? What is the effect of the output-averaging design versus a simpler alternative (e.g., a learned router)?

**Questions:**

In zero-shot evaluations, do you use generation results or likelihood? Why there is a gap between PPL performance and zero-shot performance?

---

> ### Author Response · Authors · 2025-11-17
>
> > Reasoning Claims Are Not Supported by Appropriate Evaluations The paper's most significant weakness is the disconnect between its framing and its results. The work is motivated as an advance toward "thinking" beyond Chain-of-Thought (CoT), yet it is not evaluated on any of the standard multi-step reasoning benchmarks (e.g., GSM8K, MATH, Big-Bench Hard) where such capabilities are measured. The authors acknowledge this as a limitation, but its omission is critical. Without this data, the central claim—that the "bubbles" constitute meaningful "thinking"—is unsubstantiated.
>
> To the best of our knowledge there is currently no publicly known method that enables training a model from scratch on academic budget across all ablations, especially for datasets like Big-Bench Hard. However, we do revise the language of thinking in our article, and note that it does exhibit better scaling behavior compared to a baseline GPT-2 style model that has been trained on double the amount of data.
>
> >  (1) Weak FLOPs-Matched Baseline: The "Copy-N" baseline is a poor and non-competitive comparator. It measures the effect of applying a small model to more data, not the performance of a different model with an equivalent compute budget. A far more competitive and necessary baseline is a standard, non-adaptive GPT-2-style model that is scaled up (e.g., made wider or deeper) to have the same training and/or inference FLOPs as the "Ours (k=4L)" model. It is very plausible that a standard, larger transformer would outperform this complex, adaptive one.
>
> In Figure 3 of our work, we show that our approach at the 300M scale outperforms a baseline ran at 700M scales—even though the latter requires both more compute at inference time as well as has more parameters. Furthermore, in the overall comment above, we note that our approach performs better than a baseline transformer at the same scale but using the double the training  data (and thus training compute).
>
> > (2) Missing Wall-Clock Comparison: The paper omits a practical, wall-clock time comparison. The authors admit to "low raw wall-clock efficiency" due to the overhead of forking, gating, and averaging
>
> Our method does not competitively perform against wall-clock comparisons not because of its theoretical efficiency (see above, where we perform more computation matched experiments with an upper bound on 2x the data use as well), the practical overhead of kernel launch times and data movement of operations like top-k, as well as currently the lack of a flash attention kernel. We will achieve better performance on wall-clock by virtue of a more efficient implementation such as custom kernels with fused operations.
>
> > (3) The paper fails to compare against other, more established adaptive compute methods, such as layer-skipping, early-exits, or mixture-of-depths. Without these, it's impossible to know if this parallel-forking approach is superior to adaptive-depth.
>
> For camera-ready, we can perform baselines against mixture of depth.
>
> We note, however, that the approach is fundamentally a different type of computation than the model with which we perform with our approach: mixture of depth skips existing streams for computation and hence is upper-bounded by the input token count. Furthermore, mixture of depth changes the number of parameters used, since in their work they have to allocate more layers (though only some are activated dynamically for each forward pass) in order to have a diverse mixture effect.
>
> > Key Ablations Are Missing The paper does not provide sufficient ablation studies to isolate which components of the method are responsible for the gains. For example, what is the impact of the attenuation mechanism (masking updates/attention by score) versus the forking alone? How sensitive is the model to the placement of forking layers? What is the effect of the output-averaging design versus a simpler alternative (e.g., a learned router)?
>
> We describe in the common response above an ablation study to each component of our approach that highlights that every component contributes a degree of performance gain that we observe.
>
>  > In zero-shot evaluations, do you use generation results or likelihood? Why there is a gap between PPL performance and zero-shot performance?
>
> Approaches generalize in different aspects—leading to different relative performance—but ppl remains a strong predictor of overall model performance. We use likelihood scoring for ppl, but note in section 5.1 the adaptability of our model to autoregressive scoring.

---

### Official Review · Reviewer_Anxs · 2025-11-08

**Soundness:** 2
**Presentation:** 2
**Contribution:** 1
**Rating:** 2
**Confidence:** 4

**Summary:**

Authors propose a decoder-only Transformer called Thoughtbubbles, which forks residule streams at selected layers and prunes them under certain budget. Final logits in the architecture are then a score-weighted average over surviving streams. Experiments are conducted on OpenWebText and peS2o, and results indicate better performance (lower perplexity and better zero-shot metrics) than parameter-matched baselines.

**Strengths:**

* The "thinking" process is conducted in an unsupervised and adaptive way, which is also conceptually easy to understand
* Modification on the attention mechanism is easily adaptable to modern LLM frameworks; figures are clear and informative
* Empirical results indicate improvement over baselines on certain benchmarks

**Weaknesses:**

* Reasoning tasks like GSM8k are missing, and thus it is unclear how the proposed method helps with "thinking" in practice. While authors claim it is due to the small model size, this may raise concerns on whether the proposed architecture, if applied to continual training, can maintain its capabilities
* In addition, results on physical reasoning like PIQA are mixed, this leads to further concerns on whether the architecture indeed enabled reasoning in LLMs
* The paper requires additional ablation results. In the paper authors only provide results while forking at 3,7,11 layers -- what about forking at different layers? Also what if different score initialization schemes are applied?
* Do different forks specialize in different aspects (syntax vs semantics, different possible continuations)? Figure 4 shows forks attend to their parent, but no analysis explores whether forks compute complementary versus redundant information.
* The hard top-k selection scheme may create scenarios where tokens with high cumulative scores early get dropped later, receiving little or no gradient. Thus a deeper or more frequent forking will saturate gains at a fixed block budget. I wonder if the authors have tried other stochastic selection methods, and would other methods enable a more stable and deeper forking?

**Questions:**

See above in weakness part

---

> ### Author Response · Authors · 2025-11-17
>
> > Reasoning tasks like GSM8k are missing, and thus it is unclear how the proposed method helps with "thinking" in practice. While authors claim it is due to the small model size, this may raise concerns on whether the proposed architecture, if applied to continual training, can maintain its capabilities
>
> We note that our approach does exhibit better scaling behavior compared to a baseline GPT-2 style model that has been trained on double the amount of data, and shows improved performance as model size increases (as in table 1). To the best of our knowledge there is currently no publicly known method that enables training a model from scratch on academic budget and demonstrating non-noisy performance on evaluations like GSM8k.
>
> > In addition, results on physical reasoning like PIQA are mixed, this leads to further concerns on whether the architecture indeed enabled reasoning in LLMs
>
> While PIQA performance is mixed, note that in table 1 we observe increased performance over all baselines as model size increases. Therefore, we believe that at scale our performance will become strengthened. We will train a 1B+ parameter model for camera ready, and we expect similar scale improvements.
>
> > The paper requires additional ablation results. In the paper authors only provide results while forking at 3,7,11 layers -- what about forking at different layers? Also what if different score initialization schemes are applied?
>
> We ran a minimal sweep regarding additional forking location, and described it in Appendix B. We believe that the effect of additional forking layers is constrained with a gradient bottleneck caused by hard top-k decisions. For different score initialization, we note that our formulation gives a natural initialization on all $p_{cumulative}=1$ (since they will be gradually deceasing by being multiplied to via $p_{keep, fork}$.) The calculation of $p_{keep, fork}$ uses a linear projection initialized by Xavier initialization, which is well-represented in literature.
>
> > The hard top-k selection scheme may create scenarios where tokens with high cumulative scores early get dropped later, receiving little or no gradient. Thus a deeper or more frequent forking will saturate gains at a fixed block budget. I wonder if the authors have tried other stochastic selection methods, and would other methods enable a more stable and deeper forking?
>
> Indeed, as noted here, while forking too much may create a gradient flow problem by saturating the top-k bound, a conservative amount of forking in earlier layers improves performance even beyond a computation matched (and data-upper bounded, as we show in the overall comment above) transformer. For future work, one approach that can be explored for gradient flow involves merging deleted tokens back into residual streams, leading to no bottleneck for deleted tokens.
>
> > Do different forks specialize in different aspects (syntax vs semantics, different possible continuations)? Figure 4 shows forks attend to their parent, but no analysis explores whether forks compute complementary versus redundant information.
>
> Apart from attention analysis which shows meaningful attention from parent to child, a logit lens probe shows that forks specialize in earlier layers, but become homogeneous at the final layers. This is expected because we use a weighted average of the output distributions at the final layer.

---

### Author Response · Authors · 2025-11-17
**The language of "reasoning" in the article**

We agree with the reviewer’s discussion on the word "reasoning" in our article. Though the word was not used except when contextualizing against related work, we have further unified our article around the language of “adaptive computation” and “attention allocation.” We can additionally revise the title of our article to align with this language, as in “Thoughtbubbles: Unsupervised Adaptive Parallel Computation in Latent Space”.

---

### Author Response · Authors · 2025-11-17
**More Computation Matching Experiments**

We further perform additional computation match experiments at the 150 million parameter scale. We assessed our model’s performance against a computation matched baseline by training a baseline GPT-2 style model with additional data.

| Approach    | Number of Tokens | Validation Loss |
|-------------|-------------------|------------------|
| Baseline    | 4.8B             | 3.27             |
| Ours (k=2L) | 2.4B             | 3.17             |

We see that our approach performs competitively against continued pretraining of the baseline, achieving better performance in experiments that exceed (i.e., since our approach doesn’t immediately take up the entire maximum possible block size for the entire duration of computation) both the amount of compute and tokens that we train on.
We also note that 4.8B tokens at a 150 parameter scale is certainly overtrained. From [1][2], recall that the optimal data scale follows: $C = 6ND$

Where C is the total flops, N is the number of parameters, and D is the token count. Now, figure 4 in [1] highlights that the optimal model size for flops count of 1e19 is roughly 700 million parameters, as in our experiments. Using the expression above, we obtain that the optimal data scale for this comparison is around 2.3 billion tokens. We train for 2.5 billion tokens, as follows this expression.

We note that the scaling laws of different architectures would perform differently, yet in our work we perform strongly against isoToken measurements; we further include here the performance of our approach compared to two times the token usage.

[1] Hoffman, et al., “Training Compute-Optimal Large Language Models” (2022)

[2] https://discuss.huggingface.co/t/total-flos-vs-c-6-n-d/65593

We will complete this sweep for camera ready.

---

### Author Response · Authors · 2025-11-17
**Ablating components of our approach**

We ran a series of minimal experiments with respect to different designs of our system, ablating each aspect; for camera-ready we can complete this sweep.

**Forking Rightmost Achieves Worse Performance**

| Approach                       | Number of Tokens (OpenWebText) | Validation Loss |
|-------------------------------|----------------------------------|------------------|
| Baseline                      | 4.8B                            | 3.27             |
| Ours (k=2L, logit avg.)       | 2.4B                            | 3.17             |
| Ours (k=2L, Keep Rightmost)   | 2.4B                            | 3.21             |

**Without Attention Masking, Performance Degrades Significantly**

| Approach                         | Number of Tokens (OpenWebText) | Validation Loss |
|----------------------------------|----------------------------------|------------------|
| Baseline                         | 4.8B                            | 3.27             |
| Ours (k=2L, logit avg.)          | 2.4B                            | 3.17             |
| Ours (k=2L, don’t mask attn)     | 2.4B                            | 3.90             |

We believe this is due to the fact that cumulative scores are multiplicative; without them affecting computation in some way, the multiplicative scores across layers would simply decrease throughout computation. Thus, the top-k decision is essentially random, resulting in random forks and thus significantly decreased performance.

---

### Author Response · Authors · 2025-12-02

We thank the reviewers, and in advance the ACs, for your consideration of our work.

In particular, we thank our reviewers for noting the strength of our work, including that:
- We presented an **entirely unsupervised “thinking” process for inducing adaptive computation in pretrained LLMs** which is **easily adaptable** to modern LLM frameworks
- We demonstrated **clear and consistent empirical perplexity gains** across 150M-772M scales against both baselines and non-adaptive parallel computation approaches
- We conducted **extensive analysis on the results** we have, including motivating various aspects of our design and intuitions

Reviewers have noted a few limitations, which we have addressed in rebuttals, including:

- Lack of FLOPs matched baselines, which we address in rebuttal by demonstrating that **our method (at double the block size “thinking” budget) performs better than a traditional LLM trained for double the amount of data**
- Lack of ablations to components of our design, which we address in rebuttal via ablations by demonstrating that **both the final logit averaging and attention masking components independently contributes significant gains** that we saw in our approach
- Statement of the word “reasoning”, which we address by revising our work to unify around the language “parallel computation” and “adaptive computation”

Reviewers requested experiments on GSM8k fine-tuning, which we note is difficult to conduct with meaningful, non-noisy results (i.e. >= 5% Pass@1 accuracy) at an academic scale, but look forward to exploring post-training related approaches in future work building on this foundational architecture contribution. We believe our method opens a new paradigm in enabling pre-training time adaptive computation, unifying train and test time behaviors.

We again thank the AC for their time and for consideration of our work.

---

### Meta-Review · Area_Chair_yWM3 · 2026-01-05

**Summary:**

The paper proposes Thoughtbubbles, a Transformer variant that forks and prunes residual streams per token to enable adaptive parallel computation, with final predictions formed by score-weighted logit averaging.
Reviewers generally found the idea interesting, and the perplexity gains consistently across scales. However, two concerns dominated and drove my recommendation: (i) the paper’s “thinking/reasoning” framing is not supported by the evaluations, and (ii) the experimental comparisons lack key baselines and cost analysis, making it difficult to assess the true advantage of the approach over standard or existing adaptive-compute methods.

**Reviewer Concerns:**

Addressed in rebuttal:
- Ablations: Added experiments show that attention masking and output aggregation are important components.
- Overuse of “reasoning” language: Authors acknowledge this and plan to reframe the paper around adaptive/parallel computation.

Still outstanding:
- Evaluation mismatch: Despite planned reframing, the paper provides no convincing evidence on reasoning benchmarks, leaving the original motivation insufficiently supported.
- Baselines and efficiency: Strong FLOPs- or time-matched vanilla Transformer baselines, comparisons to adaptive-compute methods (e.g., mixture-of-depths), and wall-clock/latency analysis are still missing.
- Stability of hard top-k selection: Concerns about gradient flow and saturation are acknowledged but not empirically resolved.

**Reviewer Scores:**

My estimate of score changes after discussion:
- Anxs: 2 → 2 (core concerns remain).
- nkbZ: 4 → 4 (added ablations help, but baseline gaps remain).
- 7Vnw: 2 → 2 (framing and evaluation issues persist).
- Ck1v: 4 → 4 (efficiency and scaling concerns remain).

---

### Decision · Program_Chairs · 2026-01-26

Reject